# VARIANCE-REDUCED FORWARD-REFLECTED ALGORITHMS FOR GENERALIZED EQUATIONS

## ABSTRACT

We develop two novel stochastic variance-reduction methods to approximate a solution of generalized equations applicable to both equations and inclusions. Our algorithms leverage a new combination of ideas from the forward-reflected-backward splitting method and a class of unbiased variance-reduced estimators. We construct two new stochastic estimators within this class, inspired by the well-known SVRG and SAGA estimators. These estimators significantly differ from existing approaches used in minimax and variational inequality problems. By appropriately selecting parameters, both algorithms achieve the state-of-the-art oracle complexity of $\mathcal{O}(n + n^{2/3}\epsilon^{-2})$ for obtaining an $\epsilon$-solution in terms of the operator residual norm, where $n$ represents the number of summands and $\epsilon$ signifies the desired accuracy. This complexity aligns with the best-known results in SVRG and SAGA methods for stochastic nonconvex optimization. We test our algorithms on two numerical examples and compare them with existing methods. The results demonstrate promising improvements offered by the new methods compared to their competitors.

## 1 INTRODUCTION

**Linear and nonlinear equations and inclusions** are cornerstones of computational mathematics, finding applications in diverse fields like engineering, mechanics, economics, statistics, optimization, and machine learning, see, e.g., Bauschke & Combettes (2017); Burachik & Iusem (2008); Facchinei & Pang (2003); Phelps (2009); Ryu & Yin (2022); Ryu & Boyd (2016). These problems, known as *generalized equations* (Rockafellar & Wets, 1997), are equivalent to *fixed-point problems*. The recent revolution in deep learning and generative AI has brought renewed interest to generalized equations and their special cases: minimax problems. They serve as powerful tools for handling Nash's equilibria and minimax models in generative machine learning, adversarial learning, and robust learning, see Arjovsky et al. (2017); Goodfellow et al. (2014); Madry et al. (2018); Namkoong & Duchi (2016). Notably, most problems arising from these applications are nonmonotone, nonsmooth, and large-scale. This paper develops new and simple stochastic algorithms with variance reduction for solving this class of problems, equipped with rigorous theoretical guarantees.

### 1.1 PROBLEM STATEMENT AND MOTIVATION

**[Non]linear inclusion.** The central problem studied in this paper is the following *[non]linear composite inclusion* (also called a *generalized equation* (Rockafellar & Wets, 1997)):

$$\text{Find } x^\star \in \text{dom}(\Psi) \text{ such that: } 0 \in \Psi x^\star := Gx^\star + Tx^\star, \tag{NI}$$

where $G : \mathbb{R}^p \to \mathbb{R}^p$ is a given single-valued operator, possibly nonlinear, and $T : \mathbb{R}^p \rightrightarrows 2^{\mathbb{R}^2}$ is a multivalued mapping from $\mathbb{R}^p$ to $2^{\mathbb{R}^p}$ (the set of all subsets of $\mathbb{R}^p$). Here, $\Psi := G + T$ is the sum of $G$ and $T$, and $\text{dom}(\Psi) := \text{dom}(G) \cap \text{dom}(T)$, where $\text{dom}(R)$ is the domain of $R$.

**[Non]linear equation.** If $T = 0$, then (NI) reduces to the following *[non]linear equation*:

$$\text{Find } x^\star \in \text{dom}(G) \text{ such that: } Gx^\star = 0. \tag{NE}$$

Both (NI) and (NE) are also called *root-finding problems*. Clearly, (NE) is a special case of (NI). However, under appropriate assumptions on $G$ and/or $T$ (e.g., using the resolvent of $T$), one can also transform (NI) to (NE). Let $\text{zer}(\Psi) := \{x^\star \in \text{dom}(\Psi) : 0 \in \Psi x^\star\}$ and $\text{zer}(G) := \{x^\star \in \text{dom}(G) : Gx^\star = 0\}$ be the solution sets of (NI) and (NE), respectively, which are assumed to be nonempty.

**Variational inequality problems (VIPs).** If $T(\cdot) = \mathcal{N}_{\mathcal{X}}(\cdot)$, the normal cone of a nonempty, closed, and convex set $\mathcal{X}$ in $\mathbb{R}^p$, then (NI) reduces to the following VIP as a special case:

$$\text{Find } x^\star \in \mathcal{X} \text{ such that: } \langle Gx^\star, x - x^\star \rangle \geq 0, \quad \text{for all } x \in \mathcal{X}. \tag{VIP}$$

If $T = \partial g$, the subdifferential of a convex function $g$, then (NI) reduces to a mixed VIP, denoted by MVIP. Both VIP and MVIP cover many problems in practice, including minimax problems and Nash's equilibria, see, e.g., Burachik & Iusem (2008); Facchinei & Pang (2003); Phelps (2009).

**Minimax problem.** Another important special case of (NI) (or MVIP) is the following minimax problem, which has found various applications in machine learning and robust optimization:

$$\min_{u \in \mathbb{R}^{p_1}} \max_{v \in \mathbb{R}^{p_2}} \left\{ \mathcal{L}(u, v) := \varphi(u) + \mathcal{H}(u, v) - \psi(v) \right\}, \tag{Minimax}$$

where $\mathcal{H} : \mathbb{R}^{p_1} \times \mathbb{R}^{p_2} \to \mathbb{R}$ is a smooth function, and $\varphi$ and $\psi$ are proper, closed, and convex. Let us define $x := [u, v] \in \mathbb{R}^p$ as the concatenation of $u$ and $v$ with $p := p_1 + p_2$, $Gx := [\nabla_u \mathcal{H}(u, v), -\nabla_v \mathcal{H}(u, v)]$, and $Tx := [\partial \varphi(u), \partial \psi(v)]$. Then, the optimality condition of (Minimax) is written in the form of (NI). Since (VIP), and in particular, (Minimax) are special cases of (NI), our algorithms for (NI) in the sequel can be specified to solve these problems.

**Fixed-point problem.** Problem (NE) is equivalent to the following fixed-point problem:

$$\text{Find } x^\star \in \text{dom}(F) \text{ such that: } x^\star = Fx^\star, \tag{FP}$$

where $F := \mathbb{I} - G$ with $\mathbb{I}$ being the identity operator. Since (FP) is equivalent to (NE), our algorithms for (NE) developed in this paper can also be applied to solve (FP).

**Finite-sum structure.** In this paper, we are interested in the case where $G$ is a large finite-sum:

$$Gx := \frac{1}{n} \sum_{i=1}^n G_i x, \tag{1}$$

where $G_i : \mathbb{R}^p \to \mathbb{R}^p$ are given operators for all $i \in [n] := \{1, 2, \cdots, n\}$ and $n \gg 1$. This structure often arises from machine learning, networks, distributed systems, and data science. Note that our methods developed in this paper can be extended to tackle $Gx = \mathbb{E}_{\xi \sim \mathbb{P}}\big[\mathbf{G}(x, \xi)\big]$ as the expectation of a stochastic operator $\mathbf{G}$ involving a random vector $\xi$ defined on a probability space $(\Omega, \mathbb{P}, \Sigma)$.

**Motivation.** Our work is mainly motivated by the following aspects.

*Recent applications.* Both (NE) and (NI) cover minimax problems of the form (Minimax) as special cases. The minimax problem, especially in nonconvex-nonconcave settings, has recently gained its popularity as it provides a powerful tool to model applications in generative machine learning (Arjovsky et al., 2017; Goodfellow et al., 2014), robust and distributionally robust optimization (Ben-Tal et al., 2009; Bertsimas & Caramanis, 2011; Levy et al., 2020), adversarial training (Madry et al., 2018), online optimization (Bhatia & Sridharan, 2020), and reinforcement learning (Azar et al., 2017; Zhang et al., 2021). Our work is motivated by those applications.

*Optimality certification.* Existing stochastic methods often target special cases of (NI) such as (NE) and (VIP). In addition, these methods frequently rely on a monotonicity assumption, which excludes many problems of current interest, e.g., Alacaoglu et al. (2022); Alacaoglu & Malitsky (2021); Beznosikov et al. (2023); Gorbunov et al. (2022a); Loizou et al. (2021). Furthermore, existing methods analyze convergence based on a [duality] **gap function** (Facchinei & Pang, 2003) or a **restricted gap function** (Nesterov, 2007). As discussed in Cai et al. (2023); Diakonikolas (2020), these metrics have limitations, particularly in nonmonotone settings. It is important to note that standard gap functions are not applicable to our settings due to Assumption 1.4. Regarding oracle complexity, several works, e.g., Alacaoglu & Malitsky (2021); Beznosikov et al. (2023); Gorbunov et al. (2022a); Loizou et al. (2021) claim an oracle complexity of $\mathcal{O}(n + \sqrt{n}\epsilon^{-2})$ to attain an $\epsilon$-solution, but this is measured using a restricted gap function. Again, as highlighted in Cai et al. (2023); Diakonikolas (2020), this certification does not translate to the operator residual norm and is inapplicable to nonmonotone settings. Therefore, a direct comparison between our results and these previous works is challenging due to these methodological discrepancies.

*New and simple algorithms.* Many existing stochastic methods for solving (VIP) and (NI) rely on established techniques. These include mirror-prox/averaging and extragradient-type schemes combined with the classic Robbin-Monro stochastic approximation (Robbins & Monro, 1951) (e.g.,

Cui & Shanbhag (2021); Iusem et al. (2017); Juditsky et al. (2011); Kannan & Shanbhag (2019); Kotsalis et al. (2022); Yousefian et al. (2018)). Some approaches utilize increasing mini-batch sizes for variance reduction (e.g., Iusem et al. (2017)). Recent works have explored alternative variance-reduced methods for (NI) and its special cases (e.g., Alacaoglu et al. (2022); Alacaoglu & Malitsky (2021); Bot et al. (2019); Cai et al. (2022); Davis (2022)). However, these methods primarily adapt existing optimization estimators to approximate the operator $G$ without significant differences. Our approach departs from directly approximating $G$. Instead, we construct an intermediate object $S_\gamma^k := Gx^k - \gamma Gx^{k-1}$ as a linear combination of two consecutive evaluations of $G$ (i.e. $Gx^k$ and $Gx^{k-1}$). We then develop stochastic variance-reduced estimators specifically for $S_\gamma^k$. This idea allows us to design new and simple algorithms with a single loop for solving both (NE) and (NI) where the state-of-the-art oracle complexity is achieved (*cf.* Sections 3 and 4).

## 1.2 BASIC ASSUMPTIONS

In this paper, we consider both (NE) and (NI) covered by the following basic assumptions (see Bauschke & Combettes (2017) for terminologies and concepts used in these assumptions).

**Assumption 1.1.** [*Well-definedness*] $\mathrm{zer}(\Psi)$ of (NI) and $\mathrm{zer}(G)$ of (NE) are nonempty.

**Assumption 1.2.** [*Maximal monotonicity of $T$*] $T$ in (NI) is maximally monotone on $\mathrm{dom}(T)$.

**Assumption 1.3.** [*Lipschitz continuity of $G$*] $G$ in (1) is $L$-averaged Lipschitz continuous, i.e.:

$$\tfrac{1}{n} \sum_{i=1}^n \|G_i x - G_i y\|^2 \le L^2 \|x - y\|^2, \quad \forall x, y \in \mathrm{dom}(G). \tag{2}$$

**Assumption 1.4.** [*Weak-Minty solution*] There exist a solution $x^\star \in \mathrm{zer}(\Psi)$ and $\kappa \ge 0$ such that $\langle Gx + v, x - x^\star \rangle \ge -\kappa \|Gx + v\|^2$ for all $x \in \mathrm{dom}(\Psi)$ and $v \in Tx$.

While Assumption 1.1 is basic, Assumption 1.2 guarantees the single-valued and well-definiteness of the resolvent $J_T$ of $T$. In fact, this assumption can be relaxed to some classes of nonmonotone operators $T$, but we omit this extension. The $L$-averaged Lipschitz continuity (2) is standard and has been used in most deterministic, randomized, and stochastic methods. It is slightly stronger that the $L$-Lipschitz continuity of the sum $G$. The star-co-hypomonotonicity in Assumption 1.4 is significantly different from the star-strong monotonicity used in, e.g., Kotsalis et al. (2022). In fact, Assumption 1.4 covers a class of nonmonotone operators $G$. However, if $\kappa = 0$, then $\Psi$ is just star-monotone, i.e. $\langle Gx + v, x - x^\star \rangle \ge 0$ for all $x \in \mathrm{dom}(\Psi)$.

## 1.3 CONTRIBUTION AND RELATED WORK

Our primary goal is to develop a class of stochastic variance-reduction methods to solve both (NE) and (NI), their special cases such as (VIP) and (Minimax), and equivalent problems such as (FP).

**Our contribution.** Our main contribution can be summarized as follows.

(a) We introduce a new operator $S_\gamma^k$ in (FRO) and propose a class of unbiased variance-reduced estimators $\widetilde{S}_\gamma^k$ for $S_\gamma^k$ satisfying our Definition 2.1.

(b) We construct two instances of $\widetilde{S}_\gamma^k$ by leveraging the SVRG (Johnson & Zhang, 2013) and SAGA (Defazio et al., 2014) estimators, respectively that fulfill our Definition 2.1. These estimators are also of independent interest, and can be applied to develop other methods.

(c) We develop a stochastic variance-reduced forward-reflected method (VFR) to solve (NE) which requires $\mathcal{O}(n + n^{2/3}\epsilon^{-2})$ evaluations of $G_i$ to obtain an $\epsilon$-solution of (NE).

(d) We also design a novel stochastic variance-reduced forward-reflected-backward splitting method (VFRBS) to solve (NI) that also requires $\mathcal{O}(n + n^{2/3}\epsilon^{-2})$ evaluations of $G_i$.

Let us highlight the following points of our contribution. First, our intermediate operator $S_\gamma^k$ can be viewed as a generalization of the forward-reflected-backward splitting (FRBS) operator (Malitsky & Tam, 2020) or an optimistic gradient operator (Daskalakis et al., 2018) used in the literature. However, the chosen range $\gamma \in (1/2, 1)$ excludes these classical methods from recovering as special cases of $S_\gamma^k$. Second, since our SVRG and SAGA estimators are designed specifically for $S_\gamma^k$, they differ from existing estimators in the literature, including recent works (Alacaoglu et al., 2022; Alacaoglu & Malitsky, 2021; Bot et al., 2019). Third, both proposed algorithms are single-loop and straightforward to implement. Fourth, our algorithm for nonlinear inclusions (NI) significantly differs from existing methods, including deterministic ones, due to the additional term $\gamma^{-1}(2\gamma - 1)(y^k - x^k)$. For a comprehensive survey of deterministic methods, we refer to Tran-Dinh (2023).

Fifth, our oracle complexity estimates rely on the metric $\mathbb{E}[\|Gx^k\|^2]$ or $\mathbb{E}[\|Gx^k + v^k\|^2]$ for $v^k \in Tx^k$, commonly used in nonmonotone settings. Unlike the monotone case, this metric cannot be directly converted to a gap function, see, e.g., Alacaoglu et al. (2022); Alacaoglu & Malitsky (2021). Our complexity bounds match the best known in stochastic nonconvex optimization using SAGA or SVRG without additional enhancements, e.g., utilizing a nested technique as in Zhou et al. (2018).

**Related work.** Since both theory and solution methods for solving (NE) and (NI) are ubiquitous, see, e.g., Bauschke & Combettes (2017); Burachik & Iusem (2008); Facchinei & Pang (2003); Phelps (2009); Ryu & Yin (2022); Ryu & Boyd (2016), especially under the monotonicity, we only highlight the most recent related works and a further discussion is deferred to Supp. Doc. A.

*Weak-Minty solution.* Assumption 1.4 is known as a weak-Minty solution of (NI) (in particular, of (NE)), which has been widely used in recent works, e.g., Böhm (2022); Diakonikolas et al. (2021); Lee & Kim (2021); Pethick et al. (2022); Tran-Dinh (2023a) for deterministic methods and, e.g., Lee & Kim (2021); Pethick et al. (2023); Tran-Dinh & Luo (2023) for stochastic methods. This weak-Minty solution condition is weaker than the co-hypomonotonicity (Bauschke et al., 2020), which was used earlier in proximal-point methods (Combettes & Pennanen, 2004). Diakonikolas *et al.* exploited this condition to develop an extragradient variant (called EG+) to solve (NE). Following up works include Böhm (2022); Cai & Zheng (2022); Luo & Tran-Dinh (2022); Pethick et al. (2022); Tran-Dinh (2023a). A recent survey in Tran-Dinh (2023) provides several deterministic methods that rely on this condition. This assumption covers a class of nonmonotone operators $G$ or $G + T$.

*Stochastic approximation methods.* Stochastic methods for both (NE) and (NI) and their special cases have been extensively developed, see, e.g., Juditsky et al. (2011); Kotsalis et al. (2022); Pethick et al. (2023). Several methods exploited mirror-prox and averaging techniques such as Juditsky et al. (2011); Kotsalis et al. (2022), while others relied on projection or extragradient schemes, e.g., Cui & Shanbhag (2021); Iusem et al. (2017); Kannan & Shanbhag (2019); Pethick et al. (2023); Yousefian et al. (2018). Many of these algorithms use standard Robbin-Monro stochastic approximation with fixed or increasing batch sizes. Some other works generalized the analysis to a general class of algorithms such as (Beznosikov et al., 2023; Gorbunov et al., 2022a; Loizou et al., 2021) covering both standard stochastic approximation and variance reduction algorithms.

*Variance-reduction methods.* Variance-reduction techniques have been broadly explored in optimization, where many estimators were proposed, including SAGA (Defazio et al., 2014), SVRG (Johnson & Zhang, 2013), SARAH (Nguyen et al., 2017), and Hybrid-SGD (Tran-Dinh et al., 2019; 2022), and STORM (Cutkosky & Orabona, 2019). Researchers have adopted these estimators to develop methods for (NE) and (NI). For example, Davis (2022) proposed a SAGA-type methods for (NE) under a [quasi]-strong monotonicity. The authors in Alacaoglu et al. (2022); Alacaoglu & Malitsky (2021) employed SVRG estimators and developed methods for (VIP). Other works can be found in Bot et al. (2019); Carmon et al. (2019); Chavdarova et al. (2019); Huang et al. (2022); Palaniappan & Bach (2016); Yu et al. (2022). All of these results are different from ours. Some recent works exploited Halpern's fixed-point iterations and develop corresponding variance-reduced methods, see, e.g., Cai et al. (2023; 2022). However, varying parameters or incorporating double-loop/inexact methods must be used to achieve improved theoretical oracle complexity. We believe that such approaches may be challenging to select parameters and to implement in practice.

**Notation.** We use $\mathcal{F}_k := \sigma(x^0, x^1, \cdots, x^k)$ to denote the $\sigma$-algebra generated by $x^0, \cdots, x^k$ up to the iteration $k$. $\mathbb{E}_k[\cdot] = \mathbb{E}[\cdot \mid \mathcal{F}_k]$ denotes the conditional expectation w.r.t. $\mathcal{F}_k$, and $\mathbb{E}[\cdot]$ is the total expectation. We also use $\mathcal{O}(\cdot)$ to characterize convergence rates and oracle complexity. For an operator $G$, $\mathrm{dom}(G) := \{x : Gx \neq \emptyset\}$ denotes its domain, and $J_G$ denotes its resolvent.

**Paper organization.** Section 2 introduces our operator $S_\gamma^k$ and defines a class of stochastic estimators for $S_\gamma^k$. It also constructs two instances: SVRG and SAGA, and proves their key properties. Section 3 develops an algorithm for solving (NE) and establishes its oracle complexity. Section 4 designs a new algorithm for solving (NI) and proves its oracle complexity. Section 5 presents two concrete numerical examples. Proofs and additional results are deferred to Sup. Docs. A to E.

## 2 FORWARD-REFLECTED OPERATOR AND ITS STOCHASTIC ESTIMATORS

We first introduce a new forward-reflected operator (FRO) for $G$ in (NE) and (NI). Next, we propose a class of unbiased variance-reduced estimators for FRO. Finally, we construct two instances relying

on the two well-known estimators: SVRG from Johnson & Zhang (2013) and SAGA from Defazio et al. (2014). However, any other estimator could be used if it satisfies our Definition 2.1 below.

## 2.1 Forward-reflected operator

Our methods for solving (NE) and (NI) rely on the following intermediate operator constructed from $G$ via two consecutive iterates $x^{k-1}$ and $x^k$ controlled by a parameter $\gamma \in [0, 1]$:

$$S_\gamma^k := Gx^k - \gamma Gx^{k-1}. \tag{FRO}$$

Here, $\gamma$ plays a crucial role in our methods in the sequel as $\gamma \in \left(\frac{1}{2}, 1\right)$. Clearly, if $\gamma = \frac{1}{2}$, then we can write $S_{1/2}^k = \frac{1}{2}Gx^k + \frac{1}{2}(Gx^k - Gx^{k-1}) = \frac{1}{2}[2Gx^k - Gx^{k-1}]$ used in both the forward-reflected-backward splitting (FRBS) method (Malitsky & Tam, 2020) and the optimistic gradient method (Daskalakis et al., 2018). In deterministic unconstrained settings (i.e. solving (NE)), see (Tran-Dinh, 2023), FRBS is also equivalent to Popov's past-extragradient method (Popov, 1980), reflected-forward-backward splitting algorithm (Cevher & Vũ, 2021; Malitsky, 2015), and optimistic gradient scheme (Daskalakis et al., 2018). In the deterministic constrained case, i.e. solving (NI), these methods are different. Since $\gamma \in \left(\frac{1}{2}, 1\right)$, our methods below exclude these classical schemes. However, due to a similarity pattern of (FRO) and FRBS, we still term our operator $S_\gamma^k$ by the "**forward-reflected operator**", abbreviated by FRO.

## 2.2 Stochastic unbiased variance-reduced estimators for FRO

Now, let us propose the following class of stochastic variance-reduced estimators $\widetilde{S}_\gamma^k$ of $S_\gamma^k$.

**Definition 2.1.** A stochastic estimator $\widetilde{S}_\gamma^k$ is said to be a *stochastic unbiased variance-reduced estimator* of $S_\gamma^k$ in (FRO) if there exist three constants $\rho \in (0, 1]$, $C \geq 0$ and $\hat{C} \geq 0$, and a nonnegative sequence $\{\Delta_k\}$ such that the following three conditions hold:

$$\begin{cases} \mathbb{E}_k\big[\widetilde{S}_\gamma^k\big] & = S_\gamma^k, \\ \mathbb{E}\big[\|\widetilde{S}_\gamma^k - S_\gamma^k\|^2\big] & \leq \Delta_k, \\ \Delta_k & \leq (1-\rho)\Delta_{k-1} + \frac{C}{n} \cdot \sum_{i=1}^n \mathbb{E}\big[\|G_i x^k - G_i x^{k-1}\|^2\big] \\ & \quad + \frac{\hat{C}}{n} \cdot \sum_{i=1}^n \mathbb{E}\big[\|G_i x^{k-1} - G_i x^{k-2}\|^2\big]. \end{cases} \tag{3}$$

Here, $\Delta_{-1} \geq 0$, $x^{-2} = x^{-1} = x^0$, and $\mathbb{E}_k\big[\cdot\big]$ and $\mathbb{E}\big[\cdot\big]$ are the conditional and total expectations defined earlier, respectively. The condition $\rho > 0$ is important to achieve a variance reduction as long as $x^k$ is close to $x^{k-1}$ and $x^{k-1}$ is close to $x^{k-2}$. Otherwise, $\widetilde{S}_\gamma^k$ may not be a variance-reduced estimator of $S_\gamma^k$. Since $S_\gamma^k$ is evaluated at both $x^{k-1}$ and $x^k$, our bounds for the estimator $\widetilde{S}_\gamma^k$ depends on three consecutive points $x^{k-2}$, $x^{k-1}$, and $x^k$, which is different from previous works, including Alacaoglu et al. (2021); Beznosikov et al. (2023); Davis (2022); Driggs et al. (2020).

We now construct two estimators that satisfy Definition 2.1 using SVRG (Johnson & Zhang, 2013) and SAGA (Defazio et al., 2014).

(a) **Loopless-SVRG estimator for $S_\gamma^k$.** Consider a mini-batch $\mathcal{B}_k \subseteq [n] := \{1, 2, \cdots, n\}$ with a fixed batch size $b := |\mathcal{B}_k|$. Denote $G_{\mathcal{B}_k} z := \frac{1}{b} \sum_{i \in \mathcal{B}_k} G_i z$ for a given $z \in \text{dom}(G)$. We define the following estimator for $S_\gamma^k$ in (FRO):

$$\widetilde{S}_\gamma^k := (1-\gamma)(Gw^k - G_{\mathcal{B}_k} w^k) + G_{\mathcal{B}_k} x^k - \gamma G_{\mathcal{B}_k} x^{k-1}, \tag{L-SVRG}$$

where the reference or the snapshot point $w^k$ is selected randomly as follows:

$$w^{k+1} := \begin{cases} x^k & \text{with probability } \mathbf{p} \\ w^k & \text{with probability } 1 - \mathbf{p}. \end{cases} \tag{4}$$

The probability $\mathbf{p} \in (0, 1)$ will appropriately be chosen later by flipping a coin. This estimator is known as a loopless variant (Kovalev et al., 2020) of the SVRG estimator (Johnson & Zhang, 2013). However, it is different from existing estimators used for root-finding problems, including Davis (2022) because we define it for $S_\gamma^k$, not for $Gx^k$. In addition, the first term is also damped by a factor $(1-\gamma)$ to guarantee the unbiasedness of $\widetilde{S}_\gamma^k$ to $S_\gamma^k$.

The following lemma shows that our estimator $\widetilde{S}_\gamma^k$ satisfies Definition 2.1.

**Lemma 2.1.** *Let $S_\gamma^k$ be given by* (FRO) *and $\widetilde{S}_\gamma^k$ be generated by the SVRG estimator* (L-SVRG) *and*

$$\Delta_k := \tfrac{1}{nb} \sum_{i=1}^n \mathbb{E}\big[\|G_i x^k - \gamma G_i x^{k-1} - (1-\gamma)G_i w^k\|^2\big].$$

*Then, $\widetilde{S}_\gamma^k$ satisfies Definition 2.1 with this $\{\Delta_k\}$, $\rho := \tfrac{\mathbf{p}}{2}$, $C := \tfrac{4-6\mathbf{p}+3\mathbf{p}^2}{b\mathbf{p}}$, and $\hat{C} := \tfrac{2\gamma^2(2-3\mathbf{p}+\mathbf{p}^2)}{b\mathbf{p}}$.*

(b) **SAGA estimator for the FR operator.** Let $S_\gamma^k$ be defined by (FRO) and $G_{\mathcal{B}_k}$ be a mini-batch estimator defined as in (L-SVRG), we propose the following SAGA estimator for $S_\gamma^k$:

$$\widetilde{S}_\gamma^k := \tfrac{(1-\gamma)}{n} \sum_{i=1}^n \hat{G}_i^k + \big[G_{\mathcal{B}_k} x^k - \gamma G_{\mathcal{B}_k} x^{k-1} - (1-\gamma)\hat{G}_{\mathcal{B}_k}^k\big], \tag{SAGA}$$

where $\mathcal{B}_k \subseteq [n]$ is a mini-batch of size $b$ of $[n]$, and $\hat{G}_i^k$ for $i \in [n]$ is updated as

$$\hat{G}_i^{k+1} := \begin{cases} G_i x^k & \text{if } i \in \mathcal{B}_k, \\ \hat{G}_i^k & \text{if } i \notin \mathcal{B}_k. \end{cases} \tag{5}$$

To form $\widetilde{S}_\gamma^k$, we need to store $n$ components $\hat{G}_i^k$ computed so far for $i \in [n]$ in a table $\mathcal{T}_k := [\hat{G}_1^k, \hat{G}_2^k, \cdots, \hat{G}_n^k]$ initialized at $\hat{G}_i^0 := G_i x^0$ for all $i \in [n]$. Clearly, the SAGA estimator requires significant memory to store $\mathcal{T}_k$ if $n$ and $p$ are both large. We have the following result.

**Lemma 2.2.** *Let $S_\gamma^k$ be defined by* (FRO) *and $\widetilde{S}_\gamma^k$ be generated by the SAGA estimator* (SAGA), *and*

$$\Delta_k := \tfrac{1}{nb} \sum_{i=1}^n \mathbb{E}\big[\|G_i x^k - \gamma G_i x^{k-1} - (1-\gamma)\hat{G}_i^k\|^2\big].$$

*Then, $\widetilde{S}_\gamma^k$ satisfies Definition 2.1 with this $\{\Delta_k\}$ sequence, $\rho := \tfrac{b}{2n} \in (0,1]$, $C := \tfrac{[2(n-b)(2n+b)+b^2]}{nb}$, and $\hat{C} := \tfrac{2(n-b)(2n+b)\gamma^2}{nb}$.*

We only provide two instances: (L-SVRG) and (SAGA) covered by Definition 2.1. However, we believe that similar estimators for $S_\gamma^k$ relied on, e.g., JacSketch (Gower et al., 2021) or SEGA (Hanzely et al., 2018), among others can fulfill our Definition 2.1.

## 3 A VARIANCE-REDUCED FORWARD-REFLECTED METHOD FOR (NE)

Let us first utilize the class of stochastic estimators proposed in Definition 2.1 to develop a stochastic variance-reduced forward-reflected method for solving (NE) under Assumptions 1.3 and 1.4.

### 3.1 THE VFR METHOD AND ITS CONVERGENCE GUARANTEE

(a) **Variance-reduced Forward-Reflected Method (VFR).** Our method is described as follows. *Starting from $x^0 \in \text{dom}(G)$, at each iteration $k \geq 0$, we construct an estimator $\widetilde{S}_\gamma^k$ that satisfies Definition 2.1 with parameters $\rho \in (0,1]$, $C \geq 0$, and $\hat{C} \geq 0$, and then update*

$$x^{k+1} := x^k - \eta \widetilde{S}_\gamma^k, \tag{VFR}$$

*where $\eta > 0$ and $\gamma > 0$ are determined below, $x^{-1} = x^{-2} := x^0$, and $\widetilde{S}_\gamma^0 := (1-\gamma)Gx^0$.*

There are at least two stochastic estimators $\widetilde{S}_\gamma^k$ satisfying Definition 2.1 can be used in (VFR):

- The *Loopless-SVRG estimator* $\widetilde{S}_\gamma^k$ constructed by (L-SVRG).
- The *SAGA estimator* $\widetilde{S}_\gamma^k$ constructed by (SAGA).

In terms of *per-iteration complexity*, each iteration $k$ of VFR, the loopless SVRG instance requires three mini-batch evaluations $G_{\mathcal{B}_k} w^k$, $G_{\mathcal{B}_k} x^k$, and $G_{\mathcal{B}_k} x^{k-1}$ of $G$, and occasionally computes one full evaluation $Gw^k$ of $G$ with the probability $\mathbf{p}$. It needs one more mini-batch evaluation $G_{\mathcal{B}_k} x^{k-1}$ compared to SVRG-type methods for optimization. Similarly, the SAGA instance also requires two mini-batch evaluations $G_{\mathcal{B}_k} x^k$ and $G_{\mathcal{B}_k} x^{k-1}$, which is one more mini-batch $G_{\mathcal{B}_k} x^{k-1}$ compared to SAGA-type methods in optimization, see, e.g., Reddi et al. (2016a). The SAGA estimator can avoid the occasional full-batch evaluation $Gw^k$ from L-SVRG, but as a compensation, we need to store a table $\mathcal{T}_k := [\hat{G}_1^k, \hat{G}_2^k, \cdots, \hat{G}_n^k]$, which requires significant memory in the large-scale regime.

(b) **Convergence guarantee.** Fixed $\gamma \in \left(\frac{1}{2}, 1\right)$, with $\rho$, $C$, and $\hat{C}$ as in Definition 2.1 we define

$$M := \tfrac{\gamma(1+5\gamma)}{3(2\gamma-1)} + \tfrac{1+6\gamma}{3(2\gamma-1)} \cdot \tfrac{C+\hat{C}}{\rho} \quad \text{and} \quad \delta := \tfrac{2\gamma-1}{8\sqrt{M}}. \tag{6}$$

Then, the following theorem states the convergence of (VFR), whose proof is in Supp. Doc. C.

**Theorem 3.1.** *Let us fix* $\gamma \in \left(\frac{1}{2}, 1\right)$*, and define* $M$ *and* $\delta$ *as in* (6)*. Suppose that Assumptions 1.1, 1.3, and 1.4 hold for* (NE) *for some* $\kappa \geq 0$ *such that* $L\kappa \leq \delta$*. Let* $\{x^k\}$ *be generated by* (VFR) *using a learning rate* $\eta > 0$ *such that* $\frac{8\kappa}{2\gamma-1} \leq \eta \leq \frac{1}{L\sqrt{M}}$*. Then, the following bounds hold:*

$$\begin{aligned}
\tfrac{1}{K+1} \sum_{k=0}^{K} \mathbb{E}\left[\|Gx^k\|^2\right] &\leq \tfrac{2\left(1+L^2\eta^2\right)}{\gamma(1-\gamma)\eta^2(K+1)} \cdot \|x^0 - x^\star\|^2, \\
\tfrac{(1-ML^2\eta^2)}{K+1} \sum_{k=1}^{K} \mathbb{E}\left[\|x^k - x^{k-1}\|^2\right] &\leq \tfrac{8\left(1+L^2\eta^2\right)}{3(2\gamma-1)(K+1)} \cdot \|x^0 - x^\star\|^2.
\end{aligned} \tag{7}$$

Theorem 3.1 only proves a $\mathcal{O}\left(1/K\right)$ convergence rate of both $\frac{1}{K+1} \sum_{k=0}^{K} \mathbb{E}\left[\|Gx^k\|^2\right]$ and $\frac{1}{K+1} \sum_{k=1}^{K} \mathbb{E}\left[\|x^k - x^{k-1}\|^2\right]$, but does not characterize the oracle complexity of (VFR). If we choose $\gamma := \frac{3}{4}$, then from (6), we have $M = \frac{57}{24} + \frac{11(C+\hat{C})}{3\rho}$ and $\delta = \frac{1}{16\sqrt{M}}$, which can simplify the bounds in Theorem 3.1. In addition, it allows $\kappa > 0$ such that $L\kappa \leq \delta = \mathcal{O}\left(\sqrt{\rho}\right)$, which means that $\kappa$ can be positive, but depends on $\sqrt{\rho}$. This condition allows us to cover a class of nonmonotone operators $G$, where a weak-Minty solution exists as stated in Assumption 1.4.

## 3.2 ORACLE COMPLEXITY BOUNDS OF VFR USING SVRG AND SAGA ESTIMATORS

Let us first apply Theorem 3.1 to the mini-batch SVRG estimator (L-SVRG) in Section 2. For simplicity of our presentation, we choose $\gamma := \frac{3}{4}$ and $\eta := \frac{1}{L\sqrt{M}}$, but any $\gamma \in \left(\frac{1}{2}, 1\right)$ still works.

**Corollary 3.1.** *Suppose that Assumptions 1.1, 1.3, and 1.4 hold for* (NE) *with* $\kappa \geq 0$ *as in Theorem 3.1. Let* $\{x^k\}$ *be generated by* (VFR) *using the SVRG estimator* (L-SVRG)*,* $\gamma := \frac{3}{4}$*, and* $\eta := \frac{1}{L\sqrt{M}} \geq \frac{0.1440\sqrt{b}\mathbf{p}}{L}$*, provided that* $b\mathbf{p}^2 \leq 1$*. Then, the following bound holds:*

$$\tfrac{1}{K+1} \sum_{k=0}^{K} \mathbb{E}\left[\|Gx^k\|^2\right] \leq \tfrac{526L^2R_0^2}{b\mathbf{p}^2(K+1)}, \quad \text{where} \quad R_0 := \|x^0 - x^\star\|. \tag{8}$$

*For a given* $\epsilon > 0$*, if we choose* $\mathbf{p} := n^{-1/3}$ *and* $b := \lfloor n^{2/3} \rfloor$*, then* (VFR) *requires* $\mathcal{T}_{G_i} := n + \lfloor \frac{4\Gamma L^2 R_0^2 n^{2/3}}{\epsilon^2} \rfloor$ *evaluations of* $G_i$ *to achieve* $\frac{1}{K+1} \sum_{k=0}^{K} \mathbb{E}\left[\|Gx^k\|^2\right] \leq \epsilon^2$*, where* $\Gamma := 731$*.*

Corollary 3.1 states that the oracle complexity of (VFR) is $\mathcal{O}\left(n + n^{2/3}\epsilon^{-2}\right)$, matching the one of SVRG for nonconvex optimization in, e.g., Allen-Zhu & Hazan (2016); Reddi et al. (2016b) (up to a constant). It improves by a factor $\mathcal{O}\left(n^{1/3}\right)$ compared to deterministic counterparts. This complexity is known to be the best for SVRG so far without any additional enhancement (e.g., nested techniques (Zhou et al., 2018)) even for a special case of (NE): $Gx = \nabla f(x)$ in nonconvex optimization.

Note that $\eta$ can be computed explicitly when $b$ and $\mathbf{p}$ are given. For example, if $n = 10000$, and we choose $\mathbf{p} = n^{-1/3} = 0.0464$ and $b = \lfloor n^{2/3} \rfloor = 464$, then $\eta = \frac{0.1456}{L}$. If we increase $\mathbf{p} = 0.1$, then $\eta = \frac{0.3038}{L}$. Note that, in general, we can choose any $p := \mathcal{O}\left(n^{-1/3}\right)$ and $b := \mathcal{O}\left(n^{2/3}\right)$.

Alternatively, we can apply Theorem 3.1 to the mini-batch SAGA estimator (SAGA).

**Corollary 3.2.** *Suppose that Assumptions 1.1, 1.3, and 1.4 hold for* (NE) *with* $\kappa \geq 0$ *as in Theorem 3.1. Let* $\{x^k\}$ *be generated by* (VFR) *using the SAGA estimator* (SAGA)*,* $\gamma := \frac{3}{4}$*, and* $\eta := \frac{1}{L\sqrt{M}} \geq \frac{0.1494b^{3/2}}{nL}$*, provided that* $1 \leq b \leq n^{2/3}$*. Then, the following bound holds:*

$$\tfrac{1}{K+1} \sum_{k=0}^{K} \mathbb{E}\left[\|Gx^k\|^2\right] \leq \tfrac{489L^2R_0^2}{b\mathbf{p}^2(K+1)}, \quad \text{where} \quad R_0 := \|x^0 - x^\star\|. \tag{9}$$

*Moreover, for a given* $\epsilon > 0$*, if we choose* $b := \lfloor n^{2/3} \rfloor$*, then* (VFR) *requires* $\mathcal{T}_{G_i} := n + \lfloor \frac{3\Gamma L^2 R_0^2 n^{2/3}}{\varepsilon^2} \rfloor$ *evaluations of* $G_i$ *to achieve* $\frac{1}{K+1} \sum_{k=0}^{K} \mathbb{E}\left[\|Gx^k\|^2\right] \leq \epsilon^2$*, where* $\Gamma := 2816$*.*

Similar to Corollary 3.1, the learning rate $\eta$ in Corollary 3.2 can explicitly be computed if we know $n$ and $b$. For instance, if $n = 10000$, and we choose $b = \lfloor n^{2/3} \rfloor$, then $\eta = \frac{0.1603}{L}$.

If $\kappa = 0$, i.e. $G$ reduces to a star-monotone operator, then we can choose $\gamma \in \left(\frac{1}{2}, 1\right)$ and $\eta$ as:

- For SVRG: $0 < \eta \le \frac{1}{L\sqrt{M}}$. If $\mathbf{p} = \mathcal{O}\big(n^{-1/3}\big)$ and $b = \mathcal{O}\big(n^{2/3}\big)$, then $\eta = \mathcal{O}\big(\frac{1}{L}\big)$;
- For SAGA: $0 < \eta \le \frac{1}{L\sqrt{M}}$. If $b = \mathcal{O}\big(n^{2/3}\big)$, then $\eta = \mathcal{O}\big(\frac{1}{L}\big)$.

Hitherto, the constant factor $\Gamma$ in both corollaries is still relatively large, but it can be further improved by refining our technical proofs (e.g., carefully using Young's inequality).

# 4 A NEW VARIANCE-REDUCED FRBS METHOD FOR (NI)

In this section, we develop a new stochastic variance-reduced forward-reflected-backward splitting (FRBS) method to solve (NI) under Assumptions 1.2, 1.3, and 1.4.

## 4.1 THE VARIANCE-REDUCED FRBS ALGORITHM AND ITS CONVERGENCE

(a) **The variance-reduced FRBS method (VFRBS).** Our scheme for solving (NI) is as follows. *Starting from $x^0 \in \text{dom}(\Psi)$, at each iteration $k \ge 0$, we generate an estimator $\widetilde{S}_\gamma^k$ that satisfies Definition 2.1 with $\rho \in (0, 1]$, $C \ge 0$, and $\hat{C} \ge 0$ and update*

$$x^{k+1} := x^k - \eta\widetilde{S}_\gamma^k - \eta\big(\gamma v^{k+1} - (2\gamma - 1)v^k\big), \tag{VFRBS}$$

*where $\eta > 0$ and $\gamma > 0$ are determined later, $v^k \in Tx^k$, $x^{-1} = x^{-2} := x^0$, and $\widetilde{S}_\gamma^0 := (1-\gamma)Gx^0$.*

(b) **Implementable version.** Since $v^{k+1} \in Tx^{k+1}$ appears on the right-hand side of (VFRBS), using the resolvent $J_{\gamma\eta T}(\cdot) := (\mathbb{I} + \gamma\eta T)^{-1}(\cdot)$ of $T$, we can rewrite (VFRBS) equivalently to

$$\begin{cases} y^{k+1} := x^k - \eta\widetilde{S}_\gamma^k + \frac{(2\gamma-1)}{\gamma}(y^k - x^k), \\ x^{k+1} := J_{\gamma\eta T}\big(y^{k+1}\big). \end{cases} \tag{10}$$

Here, $y^0 \in \text{dom}(\Psi)$ is given, and $x^0 = x^{-1} := J_{\gamma\eta T}(y^0)$. This is an implementable variant of (VFRBS) using the resolvent $J_{\gamma\eta T}$. Clearly, if $\gamma = \frac{1}{2}$, then (10) reduces to $x^{k+1} := J_{(\eta/2)T}\big(x^k - \eta\widetilde{S}_{1/2}^k\big)$, which can be viewed as a stochastic forward-reflected-backward splitting scheme. However, our $\gamma \in \big(\frac{1}{2}, 1\big)$, making (10) different from existing methods, even in the deterministic case.

Compared to Alacaoglu & Malitsky (2021), (10) requires only one $J_{\gamma\eta T}$ as in Alacaoglu et al. (2022), while Alacaoglu & Malitsky (2021) needs more than ones. Moreover, our estimator $\widetilde{S}_\gamma^k$ is also different from the one in Alacaoglu & Malitsky (2021). Compared to Beznosikov et al. (2023) and also Alacaoglu et al. (2022), the term $\gamma^{-1}(2\gamma - 1)(y^k - x^k)$ makes it different from SGDA in Beznosikov et al. (2023) and Alacaoglu et al. (2022), and also existing deterministic methods.

(c) **Approximate solution certification.** To certify an approximate solution of (NI), we note that its exact solution $x^\star \in \text{zer}(\Psi)$ satisfies $\|Gx^\star + v^\star\|^2 = 0$ for some $v^\star \in Tx^\star$. Therefore, if $(x^k, v^k)$ satisfies $\mathbb{E}\big[\|Gx^k + v^k\|^2\big] \le \epsilon^2$ for some $v^k \in Tx^k$, then we can say that $x^k$ is an $\epsilon$-solution of (NI). Alternatively, we can define a forward-backward splitting (FBS) residual for (NI) as $\mathcal{G}_\eta x := \eta^{-1}(x - J_\eta(x - \eta Gx))$ for any $\eta > 0$. It is well-known that $x^\star \in \text{zer}(\Psi)$ iff $\mathcal{G}_\eta x^\star = 0$. Hence, if $\mathbb{E}\big[\|\mathcal{G}_\eta x^k\|^2\big] \le \epsilon^2$, then $x^k$ is also called an $\epsilon$-solution of (NI). One can easily prove that $\|\mathcal{G}_\eta x^k\| \le \|Gx^k + v^k\|$ for any $v^k \in Tx^k$. Clearly, the former metric implies the latter one. Therefore, it is sufficient to only certify $\mathbb{E}\big[\|Gx^k + v^k\|^2\big] \le \epsilon^2$, which implies $\mathbb{E}\big[\|\mathcal{G}_\eta x^k\|^2\big] \le \epsilon^2$.

(d) **Convergence analysis.** For simplicity of our presentation, for a given $\gamma \in \big(\frac{1}{2}, 1\big)$, with $\rho$, $C$, and $\hat{C}$ in Definition 2.1, we define the following two parameters:

$$M := 4\gamma^2 + \frac{4\gamma}{1-\gamma} \cdot \frac{C+\hat{C}}{\rho} \quad \text{and} \quad \delta := \frac{\gamma(2\gamma-1)}{(3\gamma-1)\sqrt{M}}. \tag{11}$$

Then, Theorem 4.1 below states the convergence of (VFRBS), whose proof is in Supp. Doc. D.

**Theorem 4.1.** *Let us fix $\gamma \in \big(\frac{1}{2}, 1\big)$, and define $M$ and $\delta$ as in (11). Suppose that Assumptions 1.1, 1.2, 1.3, and 1.4 hold for (NI) for some $\kappa \ge 0$ such that $L\kappa < \delta$. Let $\{x^k\}$ be generated by (VFRBS) using a learning rate $\eta$ such that $\frac{(3\gamma-1)\kappa}{\gamma(2\gamma-1)} < \eta \le \frac{1}{L\sqrt{M}}$. Then, we have*

$$\begin{aligned} \frac{1}{K+1}\sum_{k=0}^K \mathbb{E}\big[\|Gx^k + v^k\|^2\big] &\le \frac{\Theta\hat{R}_0^2}{\eta^2(K+1)}, \\ \frac{(1-ML^2\eta^2)}{K+1}\sum_{k=0}^K \mathbb{E}\big[\|x^k - x^{k-1}\|^2\big] &\le \frac{4(3\gamma-1)\hat{R}_0^2}{(1-\gamma)(K+1)}, \end{aligned} \tag{12}$$

*where $\Theta := \frac{(3\gamma-1)\eta}{(1-\gamma)[\gamma(2\gamma-1)\eta-(3\gamma-1)\kappa]} > 0$ and $\hat{R}_0^2 := \|x^0 - x^\star\|^2 + \gamma^2\eta^2\|Gx^0 + v^0\|^2$.*

The bounds in Theorem 4.1 are similar to the ones in Theorem 3.1, but their proof relies on a new Lyapunov function. Note that the condition on $L\kappa$ still depends on $\rho$ as $L\kappa \leq \delta = \mathcal{O}(\sqrt{\rho})$.

## 4.2 ORACLE COMPLEXITY BOUNDS OF VFRBS USING SVRG AND SAGA ESTIMATORS

Similar to Section 3, we can apply Theorem 4.1 for the mini-batch SVRG estimator in Section 2.

**Corollary 4.1.** *Suppose that Assumptions 1.1, 1.2, 1.3, and 1.4 hold for (NI) with $\kappa \geq 0$ as in Theorem 4.1. Let $\{x^k\}$ be generated by (VFRBS) using the SVRG estimator (L-SVRG), $\gamma \in \left(\frac{1}{2}, 1\right)$, and $\eta := \frac{1}{L\sqrt{M}} \geq \frac{\sigma\sqrt{b}\mathbf{p}}{L}$ with $\sigma := \frac{\sqrt{1-\gamma}}{2\sqrt{8+\gamma+7\gamma^2}}$, provided that $b\mathbf{p}^2 \leq 1$. Then, we have*

$$\frac{1}{K+1}\sum_{k=0}^{K}\mathbb{E}\big[\|Gx^k + v^k\|^2\big] \leq \frac{\Theta L^2 \hat{R}_0^2}{\sigma^2 b\mathbf{p}^2(K+1)}, \text{ where } \hat{R}_0^2 := \|x^0 - x^\star\|^2 + \gamma^2\eta^2\|Gx^0 + v^0\|^2. \quad (13)$$

*For a given $\epsilon > 0$, if we choose $\mathbf{p} := n^{-1/3}$ and $b := \lfloor n^{2/3} \rfloor$, then (VFRBS) requires $\mathcal{T}_{G_i} := n + \left\lfloor \frac{4\Gamma L^2 \hat{R}_0^2 n^{2/3}}{\varepsilon^2} \right\rfloor$ evaluations of $G_i$ and $\mathcal{T}_T = \left\lfloor \frac{\Gamma L^2 \hat{R}_0^2}{\epsilon^2} \right\rfloor$ evaluations of $J_{\gamma\eta T}$ to achieve $\frac{1}{K+1}\sum_{k=0}^{K}\mathbb{E}\big[\|Gx^k + v^k\|^2\big] \leq \epsilon^2$, where $\Gamma := \frac{\Theta}{\sigma^2}$.*

Alternatively, we can apply Theorem 4.1 to the mini-batch SAGA estimator (SAGA) in Section 2.

**Corollary 4.2.** *Suppose that Assumptions 1.1, 1.2, 1.3, and 1.4 hold for (NI) with $\kappa \geq 0$ as in Theorem 4.1. Let $\{x^k\}$ be generated by (VFRBS) using the SAGA estimator (SAGA), $\gamma \in \left(\frac{1}{2}, 1\right)$, and $\eta := \frac{1}{L\sqrt{M}} \geq \frac{\sigma b^{3/2}}{nL}$ with $\sigma := \frac{\sqrt{1-\gamma}}{2\sqrt{\gamma(10+\gamma+7\gamma^2)}}$, provided that $1 \leq b \leq n^{2/3}$. Then, we have*

$$\frac{1}{K+1}\sum_{k=0}^{K}\mathbb{E}\big[\|Gx^k + v^k\|^2\big] \leq \frac{n^2\Theta L^2 \hat{R}_0^2}{\sigma^2 b^3(K+1)}, \text{ where } \hat{R}_0^2 := \|x^0 - x^\star\|^2 + \gamma^2\eta^2\|Gx^0 + v^0\|^2. \quad (14)$$

*For a given $\epsilon > 0$, if we choose $b := n^{2/3}$, then (VFRBS) requires $\mathcal{T}_{G_i} := n + \left\lfloor \frac{3\Gamma L^2 \hat{R}_0^2 n^{2/3}}{\varepsilon^2} \right\rfloor$ evaluations of $G_i$ and $\mathcal{T}_T = \left\lfloor \frac{\Gamma L^2 \hat{R}_0^2}{\epsilon^2} \right\rfloor$ evaluations of $J_{\gamma\eta T}$ to achieve $\frac{1}{K+1}\sum_{k=0}^{K}\mathbb{E}\big[\|Gx^k + v^k\|^2\big] \leq \epsilon^2$, where $\Gamma := \frac{\Theta}{\sigma^2}$.*

Similar to Subsection 3.2, when $\gamma$, $n$, $b$, and $\mathbf{p}$ are given, we can compute concrete values of the theoretical learning rate $\eta$ in both corollaries. They are larger than the corresponding lower bounds.

## 5 NUMERICAL EXPERIMENTS

We provide two examples to illustrate (VFR) and (VFRBS) and compare them with other methods.

**Example 1.** We consider the following unconstrained nonconvex-nonconcave minimax problem:

$$\min_{u \in \mathbb{R}^{p_1}} \max_{v \in \mathbb{R}^{p_2}} \left\{ \mathcal{L}(u, v) := \frac{1}{n}\sum_{i=1}^{n}\left[u^T A_i u + u^T L_i v - v^T B_i v + b_i^\top u - c_i^\top v\right]\right\}, \quad (15)$$

where $A_i \in \mathbb{R}^{p_1 \times p_1}$ and $B_i \in \mathbb{R}^{p_2 \times p_2}$ are symmetric matrices, $L_i \in \mathbb{R}^{p_1 \times p_2}$, $b_i \in \mathbb{R}^{p_1}$, and $c_i \in \mathbb{R}^{p_2}$. The optimality of (15) becomes **Equation** (NE) (see Supp. Doc. E for details).

We generate $A_i = Q_i D_i Q_i^T$ for a given orthonormal matrix $Q_i$ and a diagonal matrix $D_i$, where its elements $D_i^j$ are generated from standard normal distribution and clipped as $\max\{D_i^j, -0.1\}$. The matrix $B_i$ is also generated by the same way, while $L_i$, $b_i$, and $c_i$ are generated from standard normal distribution. In this case, $\mathbf{G}$ in (NE) is not symmetric and possibly not positive semidefinite.

We implement three variants of (VFR) to solve (15): `VFR-svrg` (double-loop SVRG), `LVFR-svrg` (loopless SVRG), `VFR-saga` (using SAGA estimator) in Python. We also compare our methods with the deterministic optimistic gradient method (`OG`) in Daskalakis et al. (2018), the variance-reduced FRBS scheme (VFRBS) in Alacaoglu et al. (2022), and the variance-reduced extragradient algorithm (`VEG`) in Alacaoglu & Malitsky (2021). We select the parameters as suggested by our theory, while choosing appropriate parameters for `OG`, VFRBS, and `VEG`. The details of this experiment, including generating data and specific choice of parameters, are given in Supp. Doc. E.

The relative residual norm $\|Gx^k\|/\|Gx^0\|$ against the number of epochs averaged on 10 problem instances is revealed in Figure 1 for two datasets $(p, n) = (100, 5000)$ and $(p, n) = (200, 10000)$.

Clearly, with these experiments, three SVRG variants of our method (VFRBS) work well and significantly outperform other competitors. The `LVFR-svrg` variant of (VFRBS) seems to work best, while VFRBS and `VEG` still cannot beat the deterministic algorithm `OG` in this example.

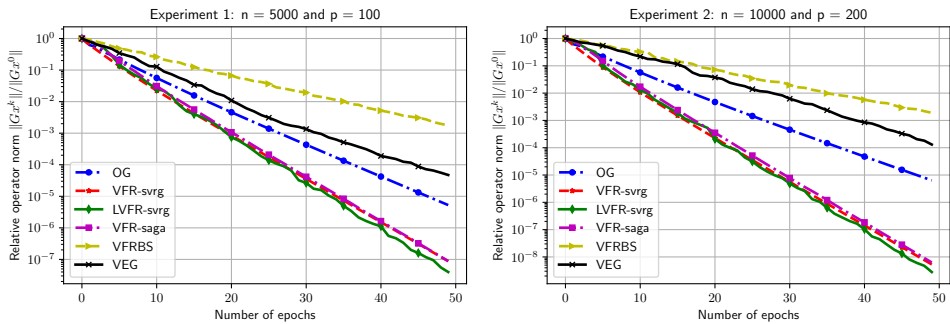

Figure 1: Comparison of 6 algorithms to solve (15) on 2 experiments (The average of 10 runs).

**Example 2.** We consider the following minimax problem arising from a regularized logistic regression with ambiguous features (see Supp. Doc. E for the details of modeling this problem):

$$\min_{w \in \mathbb{R}^d} \max_{z \in \mathbb{R}^m} \Big\{ \mathcal{L}(w, z) := \frac{1}{N} \sum_{i=1}^{N} \sum_{j=1}^{m} z_j \ell(\langle X_{ij}, w \rangle, y_i) + \tau R(w) - \delta_{\Delta_m}(z) \Big\}, \quad (16)$$

where $\ell(\tau, s) := \log(1 + \exp(\tau)) - s\tau$ is the standard logistic loss, $R(w) := \|w\|_1$ is an $\ell_1$-norm regularizer, $\tau > 0$ is a regularization parameter, and $\delta_{\Delta_m}$ is the indicator of $\Delta_m$ that handles the constraint $z \in \Delta_m$. Then, the optimality condition of (16) can be cast into (NI), where $x := [w, z]$.

We implement three variants of (VFRBS) to solve (16): VFR-svrg, LVFR-svrg, and VFR-saga. We also compare our methods with OG, VFRBS, and VEG as in **Example 1**. We cary out a fine tuning procedure to select appropriate learning rates for all methods. We test these algorithms on two real datasets: a9a (134 features and 3561 samples) and w8a (311 features and 45546 samples) downloaded from LIBSVM (Chang & Lin, 2011). We first normalize the feature vector $\hat{X}_i$ and add a column of all ones to address the bias term. To generate ambiguous features, we take the nominal feature vector $\hat{X}_i$ and add a random noise generated from a normal distribution of zero mean and variance of $\sigma^2 = 0.5$. In our test, we choose $\tau := 10^{-3}$ and $m := 10$. The relative FBS residual norm $\|\mathcal{G}_\eta x^k\|/\|\mathcal{G}_\eta x^0\|$ against the epochs is plotted in Figure 2 for both datasets.

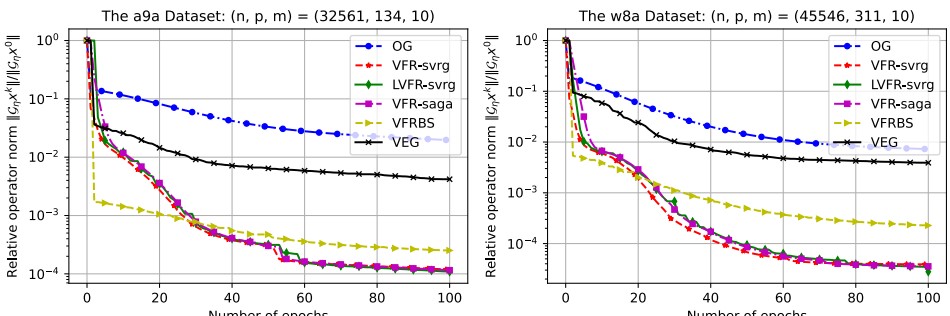

Figure 2: Comparison of 6 algorithms to solve (16) on two real datasets: a8a and w8a.

As we can observe from Figure 2 that three variants VFR-svrg, LVFR-svrg, and VFR-saga have similar performance and are better than their competitors. Among three competitors, VFRBS still works well, and is much better than OG and VEG. The deterministic method, OG, is the worst one in terms of oracle complexity. In this test, VEG has a larger learning rate than ours and VFRBS.

## 6 CONCLUSIONS

This work introduces two innovative variance-reduced algorithms based on the forward-reflected-backward splitting method to tackle equations (NE) and inclusions (NI). These methods encompass both SVRG and SAGA estimators as special cases. By carefully selecting the parameters, our algorithms achieve the state-of-the-art oracle complexity for reaching an $\epsilon$-solution, matching the state-of-the-art complexity bounds observed in nonconvex optimization methods using SVRG and SAGA. While the first scheme resembles a stochastic variant of the optimistic gradient method, the second algorithm is entirely novel and distinct from existing approaches, even their deterministic counter-parts. We have validated our methods through numerical examples, and the results demonstrate promising performance compared to existing techniques under carefully tuned parameter selections.

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
