# OpenReview forum: "Variance-Reduced Forward-Reflected Algorithms for  Generalized Equations"
_ICLR.cc/2025/Conference — ICLR 2025 Conference Withdrawn Submission_

### Official Review · Reviewer_kZ6e · 2024-10-15

**Soundness:** 1
**Presentation:** 2
**Contribution:** 1
**Rating:** 3
**Confidence:** 3

**Summary:**

This paper proposes SVRG/SAGA-type methods for generalized equations with the $O(n+n^{2/3} \epsilon^{-2})$ complexity. But the rates in this paper seems not to be competitive with related works.

**Strengths:**

I think the paper is clearly written.

**Weaknesses:**

The authors said, "Alacaoglu & Malitsky (2021); Beznosikov et al. (2023); Gorbunov
et al. (2022a); Loizou et al. (2021) claim an oracle complexity of $O(n+ \sqrt{n} \epsilon^{-2})$ to attain an $\epsilon$
solution, but this is measured using a restricted gap function. Again, as highlighted in Cai et al.
(2023); Diakonikolas (2020), this certification does not translate to the operator residual norm and is
inapplicable to nonmonotone settings."
However, it seems that [1] has proved the result of $O(n+ \sqrt{n } \epsilon^{-1})$ using SVRG-type algorithm for operator residual norm. How could the results in this work be competitive in that in [1]?

The authors said "However, varying parameters or incorporating double loop/inexact methods in [1] must be used to achieve improved theoretical oracle complexity. We believe
that such approaches may be challenging to select parameters and to implement in practice."
However, the experiments in this paper are quite weak (a synthetic experiment and a real-world experiment on logistic regression). If the proposed methods have a worse dependency on both $n$ and $\epsilon$, but the authors still claim their methods are practical, maybe they should conduct a lot of large-scale experiments to support their claim, such as  GAN and adversarial training in neural networks.

[1] Cai, Xufeng, Ahmet Alacaoglu, and Jelena Diakonikolas. "Variance Reduced Halpern Iteration for Finite-Sum Monotone Inclusions." The Twelfth International Conference on Learning Representations.

**Questions:**

1. Can the authors clarify how their $O(n+n^{2/3}\epsilon^{-2})$ complexity compares to the $O(n+\sqrt{n}\epsilon^{-1})$ complexity in [1] for the operator residual norm?

2. What are the key differences in assumptions or problem settings between this work and [1] that lead to the different complexity results?

[1] Cai, Xufeng, Ahmet Alacaoglu, and Jelena Diakonikolas. "Variance Reduced Halpern Iteration for Finite-Sum Monotone Inclusions." The Twelfth International Conference on Learning Representations.

---

### Official Review · Reviewer_LMEQ · 2024-10-29

**Soundness:** 3
**Presentation:** 2
**Contribution:** 2
**Rating:** 5
**Confidence:** 3

**Summary:**

This paper considers variance reduction methods for inclusion problems satisfying the (nonmonotone) weak Minty variational inequality (MVI) which is parameterized by some parameter $\kappa$, whose magnitude controls the allows level of nonmonotonicity.
Existing methods can largely be divided into two groups ($n$ is the number of elements in the finite sum and $\varepsilon$ is the solution quality):

- Halpern based method, which achieves a $\tilde{\mathcal{O}}(n+\sqrt{n}\varepsilon^{-1})$ complexity for making the norm of the operator small for the last iterate (see e.g. Cai et al. 2023 developing for cohypomonotone problems)
- Extragradient and single-call variants which has a ${\mathcal{O}}(n+\sqrt{n}\varepsilon^{-2})$ complexity to make the restricted gap function small for the averaged iterate (e.g. Alacaoglu & Malitsky 2021 for monotone problems)

In comparison this paper shows a ${\mathcal{O}}(n+n^{2/3}\varepsilon^{-2})$ complexity for making the norm of the operator small for the best iterate in weak MVIs. The paper shows convergence using the Loopless-SVRG and SAGA estimator combined with a single-call method that additionally only uses a single projection/resolvent.

**Strengths:**

- The paper is technically solid, treats variance reduction methods for weak MVI thoroughly, and does so in a modular fashion
- The algorithmic construction in the constrained case (eq. 10) seems new and might be interesting in its own right (see question)

**Weaknesses:**

It seems like there are primarily two contributions over variance reduction methods for monotone problems regarding the complexity:

- The guarantee is provided in terms of the norm of the operator (of the best iterate) instead of the gap (of the averaged iterate).
- The guarantees extend to some range of weak MVIs (so beyond monotone problems)

**W.1** Considering this, the main concern is that the allowed range of the parameter $\kappa$ controlling the nonmonontonicity can be restrictive and that this is not discussed in sufficient detail. Take for instance Thm. 3.1 where $\kappa \leq \mathcal O(\sqrt{\rho}/L)$:

- In the case of SAGA we have $\rho =b/(2n) \leq 1/(2n^{1/3})$ (Cor. C.2)
- in the case of SVRG we have $\rho =p/2 =1/(2n^{1/3})$ (Cor. C.1)

In either case $\kappa \leq \mathcal O (1/(Ln^{1/6}))$, so the nonmonotonicity parameter scales inversely with $n$. Additionally, the Lipschitz-averaged constant can be much larger than the Lipschitz constant ($n$-dependent). I found it very hard to parse whether there are any other dependencies on $n$, through e.g. the stepsize requirement ($\eta$) in Thm 3.1, which depends on $C$ defined in Def. 2.1, while explicitly stated in Lem. 2.1, which contains again $\rho$ through $p$. It would be very helpful if the authors could explicitly spell out the dependency. The range of $\kappa$ is particularly important considering that even extragradient and optimistic methods (without relaxation) can achieve some range of $\kappa$ (see e.g. [1] albeit for cohypomonotone problems).

 **W.2** The FRO estimator (l. 222)  seems to be equivalent to what is used for the optimistic method in Böhm 2022, so I maybe wouldn't consider this a new construction. The construction in the constrained case (eq. 10), seems new, but in the current presentation its role is unclear. Is additionally term needed even in the monotone case or is it possible to get convergence for the norm of the operator already with FBF/FoRB based methods ala Alacaoglu & Malitsky 2021 (i.e. with $\gamma=1/2$ in the current paper)?

 [1]: https://arxiv.org/abs/2210.13831

**Questions:**

- What is the dependency on the number of elements $n$ for the nonmonotonicity parameter $\kappa$?
- The construction eq. 10 seems interesting even in the deterministic case. What range of $\kappa$ does the method achieve for $n=1$?

---

### Official Review · Reviewer_62Yw · 2024-11-03

**Soundness:** 3
**Presentation:** 2
**Contribution:** 2
**Rating:** 5
**Confidence:** 2

**Summary:**

This paper studies nonlinear inclusions with the form $G x + T x$, where $G$ has the finite-sum structure of $G= \frac{1}{n}\sum_{i=1}^nG_i$. The authors use several standard variance reduction technique to reduce the number of oracle calls $G_i$ and establish the complexity of $\mathcal{O}(n+n^{2/3}\epsilon^{-2})$.

**Strengths:**

1. The paper studies a very general problem, which covers lots of important optimization problems like nonlinear equations, variational inequalities, minimax problems.

2. The author propose variance reduction version of foward-reflected method and establish new state-of-the-art oracle complexities for this problem.

**Weaknesses:**

1. The paper is too dense and the presentation makes the readers hard to follow. For example, in Section 1.1, the finite-sum structure  should not be parallel to the problem formulation (NI, NE, VIP, Minimax).

2. In the optimality certification part, the authors claim that existing stochastic methods often target special cases of NI and a better oracle complexity of $\mathcal{O}(n+\sqrt{n}\epsilon^{-2}) is measured using a restricted gap function, which can not be directly compared. However, it is not clear what is the previous state-of-the-art results for this problem. What are the previous results for NI, NE and VIP?

3. The authors use SAGA and SVRG for the variance reduced estimator, which may lead to worse oracle complexity under the cases, why not consider SPIDER?

**Questions:**

Please refer to weakness part.

---

### Official Review · Reviewer_Q4Da · 2024-11-04

**Soundness:** 3
**Presentation:** 2
**Contribution:** 2
**Rating:** 5
**Confidence:** 3

**Summary:**

This paper proposed variance-reduced forward-reflected method for generalized equations, achieving the oracle complexity of $O(n+n^{2/3}\epsilon^{-2})$ to obtain an $\epsilon$-solution. The main idea is using SVRG and SAGA estimator, which is popular in nonconvex optimization. The experiments also show the advantage of proposed methods.

**Strengths:**

see summary

**Weaknesses:**

see questions

**Questions:**

1. The maximally monotone mentioned in Assumption 1.2 should be explicitly defined (like equation (2) in Assumption 1.3).
2. Can you provide some comparison for Assumption 1.4 with other nonmonotone operators (e.g., negative comonotone and interaction dominate (Lee & Kim, 2021))?
3. It is better to provide a table to compare the complexity of proposed methods with baseline.
4. Is it possible to apply SPIDER/SARAH estimator to reduce the complexity from $O(n+n^{2/3}\epsilon^{-2})$ to $O(n+n^{1/2}\epsilon^{-2})$?
5. For Example 1 (equation (15)), the matrix $A_i$ and $B_i$ may be indefinite. How to guarantee the minimax problem be well-defined?

---

### Note · Authors · 2024-11-19

I have read and agree with the venue's withdrawal policy on behalf of myself and my co-authors.